# Using Rashomon Sets for Robust Active Learning

**Simon Dovan Nguyen**
Department of Statistics
University of Washington
Seattle, WA 98101
simondn@uw.edu

**Kentaro Hoffman**
Department of Statistics
University of Washington
Seattle, WA 98101
khoffm3@uw.edu

**Tyler H. McCormick**
Department of Statistics
University of Washington
Seattle, WA 98101
tylermc@uw.edu

## Abstract

Active learning is based on selecting informative data points to enhance model predictions, often using uncertainty as a selection criterion. However, when ensemble models such as random forests are used, there is a risk of the ensemble containing models with poor predictive accuracy or duplicates with the same interpretation. To address this, we develop a novel approach to only ensemble the set of near-optimal models called the Rashomon set in order to guide the active learning process. We demonstrate how taking a Rashomon approach can improve not only the accuracy and rate of convergence of the active learning procedure, but can also lead to improved interpretability compared to traditional approaches.

## 1 Introduction

Collecting labeled data to train data-hungry modern artificial intelligence (AI) and machine learning (ML) models can be expensive or time-consuming. This challenge arises in a wide range of applications: sentence classification [17], image labelling [23] [10], and verbal autopsy [4] . In such scenarios, strategically determining which observations merit labeling will greatly reduce data redundancy and improve the learning of covariate-label relationships.

To address time and budget constraints, active learning allows researchers the freedom to adaptively choose which observations to label. The key task in active learning is choosing the most informative observations that will enhance the predictive quality of the model when labelled. Amongst the many metrics of informativity [12] [5] [15], uncertainty is the most commonly employed [13].

Due to their ease in measuring uncertainty, ensemble techniques such as random forests are a popular model used for active learning [14]. Since the weak learners of ensemble methods are independent by design, the individual base learners naturally form a committee. When used in active learning, the disagreement in "votes" between the ensemble committee members is often used as a measure of uncertainty and informativity [11] [3].

While ensemble methods offer a natural way to quantify uncertainty through the random diversity of their weak learners, this diversity comes with a potential drawback. Specifically, most ensemble methods tend to aggregate over the space of all models, even if some of the models may have relatively poor accuracy. While some approaches such as Bayesian model averaging account for this by weighting the models by how likely they are given the data, having a large number of mediocre models are known to make such weighting approaches difficult, especially in cases of limited, noisy, or high-dimensional data [6]. Aggregating such poor and implausible models compromises the query-selection criteria, potentially leading to a suboptimal query in the active learning process.

To address this limitation, we propose a novel approach to improving the quality of ensemble methods used in active learning. Specifically, we propose an algorithm that enhances active learning with random forests by restricting aggregation to a subset of well-performing, high-evidence models known as the Rashomon set. The Rashomon set consists of near-optimal models that have strong

38th Conference on Neural Information Processing Systems (NeurIPS 2024).

support from the observed data. By ensembling only models within the Rashomon set, our approach ensures that the active learning process is driven by models with high evidence, leading to better query-selection criteria and improved query-selection.

The main results illustrating the benefits of aggregating across the Rashomon set in ensemble learning can be seen in Figure 2, in which the TreeFarms approach (blue and orange lines, whose distinction will become clear later) consistently outperforms the traditional random forest approach. We also demonstrate that one can further restrict the Rashomon set to only select models with similar "explanations" while still preserving the performance of the active learning process. This Rashomon-based method ensures that the ensemble incorporates the interpretability of the weak learners in our ensemble while maintaining prediction accuracy.

## 2 Rashomon Sets

When constructing machine learning models, researchers face two district types of uncertainty. The first originates from the variability in the predicted outcomes generated by a given model, often referred to as a model's intrinsic risk. The second originates from selecting the right model from a vast and diverse hypothesis space, a phenomenon known as *model ambiguity*. This distinction, originally articulated by economist Kenneth Arrow in 1951, separates the uncertainty of prediction from given a model from the uncertainty of choosing among many plausible models [1]. In his 2013 Nobel lecture, Lars Hansen further highlights this idea by characterizing the distinction as "uncertainty outside and inside [economic] models" [7].

Current machine learning approaches have become exceedingly good at reducing the predictive uncertainty within a model, but often fail to fully account for model ambiguity. Methods such as LASSO search for a single optimal model while ensemble methods such as Bayesian Model Averaging [9] sample across the full hypothesis space. However, both approaches overlook model ambiguity and are ambivalent to how many models with similar predictive power exist for a given dataset [16]. This oversight is underscored by the Rashomon Effect, first noted by Leo Breiman in 2001 [2]. The Rashomon Effect highlights the existence of near-optimal models that have similarly high predictive performance, but explain the data in different, potentially conflicting ways. Rudin furthers this idea by noting that this phenomenon exposes a core issue in the current machine learning paradigm: a reliance on a single predictive model that is overly-sensitive [16] [18]. This reliance fails to notice the complexity of modeling heterogeneity, where different models can explain the data nearly equally well but offer substantively different insights.

To quantify the number of near-equivalent models exist for a given dataset, one can employ techniques from Rashomon Theory. Rashomon Theory is focused on the Rashomon sets — a collection of models that are all near-optimal in terms of predictive accuracy. By enumerating the Rashomon set, researchers can explore the full range of plausible explanations supported by the data. Traditional ensemble methods on the other hand such as random forests aggregate base learners based on the random sampling of features and data. This leads to diverse but potentially suboptimal ensembles due to the inclusion of implausible models with no way of removing such poor models. In contrast, Rashomon sets allow for the targeted aggregation of models that are only high-performing, reducing the risk of incorporating poor models in the query-selection process.

In the space of decision trees, Xin et al. is the first to provide an algorithm that completely enumerate the Rashomon set for sparse decision trees [22]. Their algorithm, `TreeFarms`, provides an exhaustive yet computationally feasible method to generate, store, and view the entire Rashomon set of decision trees. However, due to the inherent structure and geometry of decision trees, many trees in this set may offer redundant explanations of the data. This can be seen in Figure 1 and more deeply in Figure 3 in the appendix. As such, duplicate trees in the ensemble method have the potential to further skew our metric of uncertainty in the committee by artificially inflating agreement in votes.

To address this limitation, section 4 will propose a method to group trees based on their unique explanations of the data and select a representative from each group. We will then show how ensembling the Rashomon set to account for model ambiguity in the active learning process will improve our query-selection criteria.

# 3 Active Learning

## 3.1 Notation

Borrowing notation from Liu et. al (2022) [13], let observation $i$ be composed of data $(\mathbf{x}_i, y_i)$ for vector $\mathbf{x}_i$ in covariate space $\mathcal{X}$ and label $y_i$ in output space $\mathcal{Y}$. The data is sent through a supervised learning model $F(\cdot) : \mathcal{X} \to \mathcal{Y}$. When $F(\cdot)$ is an ensemble method consisting of base learners, denote the base learners at $\{f_m\}_{m=1}^M$. The model is learned from a training dataset $D_{tr} = \{(\mathbf{x}_i, y_i)\}_{i=1}^I$ and tested by an independent dataset $D_{ts} = \{(\mathbf{x}_j, y_j)\}_{j=1}^J$. The goal is to train $F(\cdot)$ to predict the labels of the out-of-sample test set with a budget-constrained number of labeled observations.

Active learning seeks to adaptively and strategically choose which unlabelled observations should be queried for oracle labeling and to then be used in the supervised learning model. Let the query iteration in the active learning framework be denoted by $n$. Denote the reservoir of unlabelled candidate observations as $D_{cdd}^{(n)} = \{(\mathbf{x}_k, y_k)\}_{k=1}^K$ with $y_k$ initially unknown. A selector $S(\cdot)$ is the strategy used to select samples from $D_{cdd}^{(n)}$ to be oracle labelled. At each iteration, $S(\cdot)$ will sample a subset of observations, denoted $B^{(n)}$, from the candidate dataset $D_{cdd}^{(n)}$ without replacement to query for oracle labeling. $B^{(n)}$ is then added to the training set and removed from the candidate set: $D_{cdd}^{(n+1)} = D_{cdd}^{(n)} \cup B^{(n)}$ and $D_{cdd}^{(n+1)} = D_{cdd}^{(n)} \backslash B^{(n)}$. The model is then retrained on the new training set as $F^{(n+1)}\left(D_{tr}^{(n+1)}\right)$. As such, the $B^{(n)}$ is chosen so as to find the observations that are most informative to improving predictive performance.

The process is repeated, gradually expanding the training set with informative observations, until the labelling budget is reached or a desired classification metric threshold is met.

## 3.2 Query-By-Committee Metrics

Picking a selector metric is a key topic in the active learning literature. Common methods are uncertainty [12], Query-By-Committee metrics [5] [19], or expected error [15]. Due to the ensembling nature of our methods, we choose to measure informativity by Query-By-Committee (QBC) metrics, particularly Argamomn-Engelson and Dagan's vote entropy [3]:

$$\delta(y, \mathbf{x}, \mathcal{C}) = \max_{\mathbf{x}} - \sum_{y \in \mathcal{Y}} \frac{\text{vote}_{\mathcal{C}}(y, \mathbf{x})}{|\mathcal{C}|} \log \frac{\text{vote}_{\mathcal{C}}(y, \mathbf{x})}{|\mathcal{C}|} \tag{1}$$

where $\text{vote}_{\mathcal{C}}(y, \mathbf{x}) = \sum_{c \in \mathcal{C}} \mathbb{I}\{c(\mathbf{x}) = y\}$ is the number of "votes" that label $y \in \mathcal{Y}$ receives for $\mathbf{x}$ amongst the members $c$ of committee $\mathcal{C}$. This selector metric is a committee-based generalization of uncertainty measures that considers the confidence of each committee member and is essentially a Bayesian adaptation of Shannon's 1948 uncertainty sampling entropy [20]. One can observe, from Equation 1 that ensembling duplicate models has the potential to overinflate the vote entropy [14] with trees from the best performing explanation group.

# 4 Algorithm/Methods

In our proposed work, the committee $\mathcal{C}$ in Equation 1 is the Rashomon set of decision trees $\mathcal{R}$. For predictor $F$, we construct a Rashomon set $\mathcal{R}$ of "near-equal" decision trees defined as the set of models whose objective function is within $\epsilon$ of the overall best model given the data. Since each near-equal model in the Rashomon set will describe the data differently, conflicting prediction labels/probabilities will arise amongst models in the Rashomon set.

To enumerate the Rashomon set of decision trees, we use Xin et al.'s TreeFarms approach [22]. TreeFarms exhaustively enumerates the Rashomon set of decision trees, allowing us to aggregate the best models in our ensemble method. However, unlike random forests, TreeFarms lacks the random sampling of features and data, making the models in TreeFarms correlated. This correlation in decision trees is a significant limitation, as correlation amongst committee members may both artificially inflate agreement in the vote and complicate interpretability [14]. To address this issue, we reduce the redundancy in TreeFarms by grouping trees based on their unique explanation of the data and selecting a single representative tree from each of these groups to ensemble. This ensures that

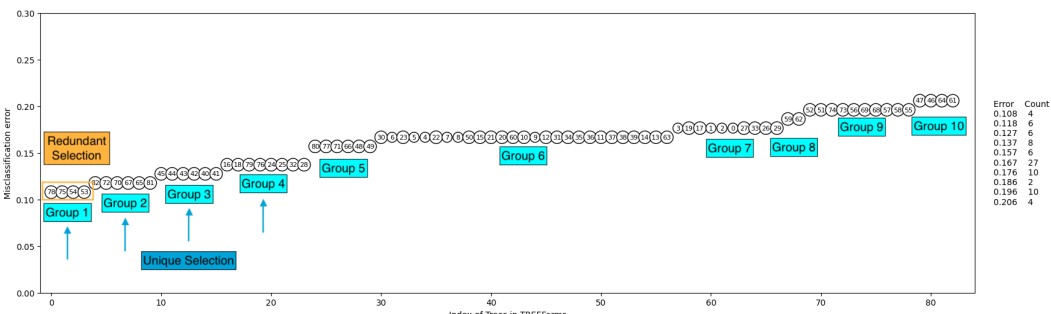

Figure 1: A depiction of how to ensemble Rashomon trees: redundantly and uniquely. This plot shows the classification error by the ordered indices of the tree. As shown, many trees have the *exact same* misclassification rate. For instance, the top 4 trees (whose geometry can be seen in Figure 3 of the appendix) share a misclassificaiton rate of 0.108. If this redundancy in the trees is not accounted for, committee agreement will be overinflated and dominated by the best performing group.

each chosen tree is meaningfully distinct while faithfully representing the Rashomon set, ultimately leading to a valid query-by-committee voting approach.

Our approach can be visualized in Figure 1. Suppose we want to define the committee of vote entropy by ensembling the top four decision trees of TreeFarms. If we ignore the the redundancy of explanations in TreeFarms, then trees 53, 54, 75, and 78 will be ensembled despite offering the same explanation and prediction. This, as noted in Equation 1, will artificially inflate agreement amongst our committee by own ensembling the trees in the top explanation groups. If we instead account for the redundancy of the trees, the unique selection method will instead choose one tree arbitrarily from Groups 1, 2, 3, and 4, diversifying our committee and more fully representing the Rashomon set. Our method is summarized in Algorithm 1.

---

**Algorithm 1:** Unique Tree Farms Active Learning

**Input :** $D_{tr}^{(0)}$; $D_{ts}$; $D_{cdd}^{(0)}$; $\epsilon$;

1 **repeat**

2      Train **F** on $D_{tr}^{(n)}$;

3      Test **F** on $D_{ts}^{(n)}$;

4      Enumerate the Rashomon set $\mathcal{R}$ of predictor $F$ with TreeFarms;

5      *(Optionally)* Reduce the Rashomon set $\mathcal{R}$ to the top $k$ models in $\mathcal{R}$;

6      Predict labels $\hat{y}_{tr,m}^{(n)}$ and calculate the classification error for each tree $f_m$ in $\mathcal{R}$;

7      Define the the smallest classification error from the $\mathcal{R}$ as the current iteration error;

8      Compute the vote-entropy metric $\delta^{(n)}(y, x, \mathcal{C})$ from equation 1 with $\mathcal{R}$ as the committee;

9      Resample $B^{(n)}$ from $D_{cdd}^{(n)}$ based on the observation with the highest vote entropy:
         $B^{(n)} := \arg\max_x \delta^{(n)}(y, x, \mathcal{C})$;

10     Query $B^{(n)}$ for oracle labeling;

11     Set $D_{tr}^{(n+1)} = D_{tr}^{(n)} \cup B^{(n)}$ and $D_{cdd}^{(n+1)} = D_{cdd}^{(n)} \setminus B^{(n)}$;

12 **until** *labelling budget is depleted or test error is sufficiently small*;

---

## 5 Experiments

One hundred active learning simulations were ran and averaged on the 1978 Boston Housing dataset of Harrison and Rubinfeld [8]. The goal was to classify whether the median value of a home was in the top 25% quantile based on five covariates: capita crime rate per town, nitric oxides concentration, average number of rooms per household, pupil-teacher ratio by town, and percent of lower status of the population. Due to the structure of TreeFarms, we discretized the five covariates into three categories (low, medium, and high) and one-hot encoded the variables. This resulted in the covariates being encoded in 15 binary variables. Open source code for the simulation is available on Github.

We compared the active learning process ensembling the top ten and 100 trees with unique explanations from the Rashomon set to its counterpart without considering the redundancy in tree explanations. At each iteration, TreeFarms was refitted with a regularization penalty on splits of 0.01 and a Rashomon $\epsilon$ of 0.05. Results can be seen in Figure 2. Figure 4 of the appendix shows the same simulation results without the baseline random forest.

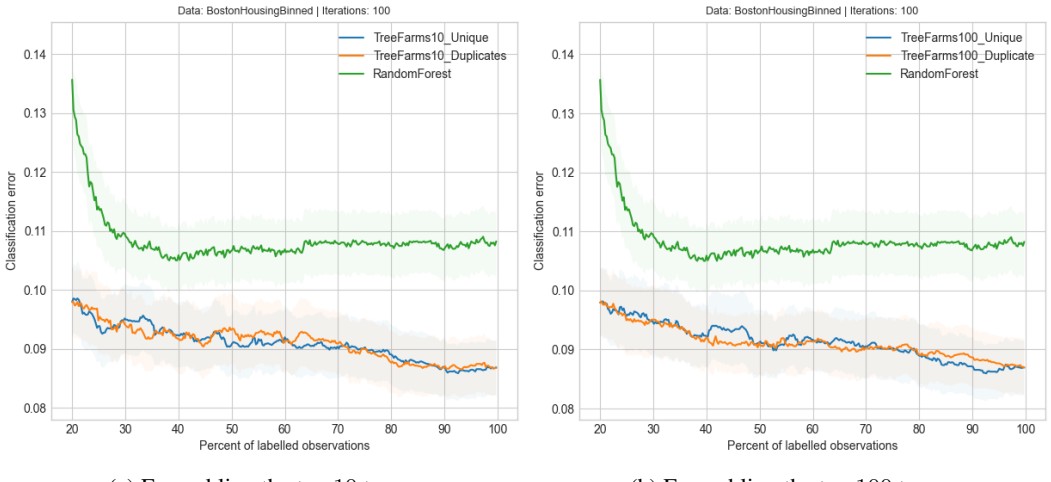

(a) Ensembling the top 10 trees.     (b) Ensembling the top 100 trees.

Figure 2: This plot gives the classification error comparing three ensemble methods: TreeFarms selecting redundant trees, TreeFarms selecting unique trees, and random forests.

Our findings demonstrate the remarkable predictive power of the Rashomon set. Ensembles of decision trees from the Rashomon set (both unique and duplicate) significantly outperformed random forests, highlighting the robustness and prediction accuracy achievable with this approach. Furthermore, removing redundant explanations in the Rashomon set by only ensembling trees with unique explanations of the data maintained classification accuracy.

This result has profound implications for interpretability. While redundancy in explanations can hinder the interpretability of an ensemble, the Rashomon framework allows us to overcome this challenge by selecting a smaller, coherent set of unique trees while maintaining prediction accuracy. This approach combines the robustness of random forests with the interpretability of individual trees

## 6    Concluding thoughts and future work

Our work offers two key insights. Firstly, we demonstrate that ensembling over the Rashomon set of decision trees enhances the active learning process by a significant margin. Unlike traditional ensemble methods which aggregate over the entire space of models, potentially including models that are poor performing or implausible, the Rashomon set only contains models with high posterior probability. This allows active learning processes to form a commmitee with only the strong and plausible models whose disagreements will then provide a more robust measure of uncertainty for more efficient query selection.

Secondly, we address the issue of redundant and duplicate explanations when constructing a Rashomon set by only considering trees with unique explanations. Redundant explanations can inflate query-by-committee metrics and obscure interpretability. By only ensembling over the Rashomon's subset of trees with unique explanations, we ensure that the ensemble remains parsimonious and interpretable while maintaining prediction accuracy.

This work *plants the seeds* for future research into other methods that form the Rashomon set without an inherent geometric structure. In particular, Rashomon Partition Sets (RPS) [21] offer a promising framework for comprehensively enumerating the Rashomon set. Investigating the use of RPS in active learning may further deepen our understanding of the Rashomon's benefits in both prediction and interpretability.

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

# 7 Appendix

## 7.1 Geometry of trees in Group 1

The images in Figure 3 give insight into the decision rules of the top four decision trees. Note that feature 0 represents whether an observation is within an area with lowest levels of crime (bottom 33% quantile), and feature 9 represents an observation in an area with the lowest pup-teacher ratio (bottom 33% quantile). Features 6, 7, and 8 represent whether an observation is in one of the three categories of rooms per household respectively: low, medium, and high.

As seen, the trees exhibit very similar decision paths to each other, resulting in each one having the *exact* same misclassification error of 0.108. As described in the main corpus, ensembling these four trees as a committee and calculating the vote entropy metric off this committee will result in an inflated agreement and will recommend the observation that the best decision tree is most uncertain of rather than consider the uncertainty of the ensemble as a whole.

## 7.2 Experimental results plotted with random forests

Figure (4) gives the classification error comparing the two ensemble methods: TreeFarms selecting redundant trees and its counterpart with only unique trees It is the same plot as Figure 2 but removes the classification error line for random forests to better visualize the differences between the inclusion of redundant vs. duplicate trees in the ensemble method. The value given in each subfigure represents the p-value from a Wilcoxon ranked-sign test.

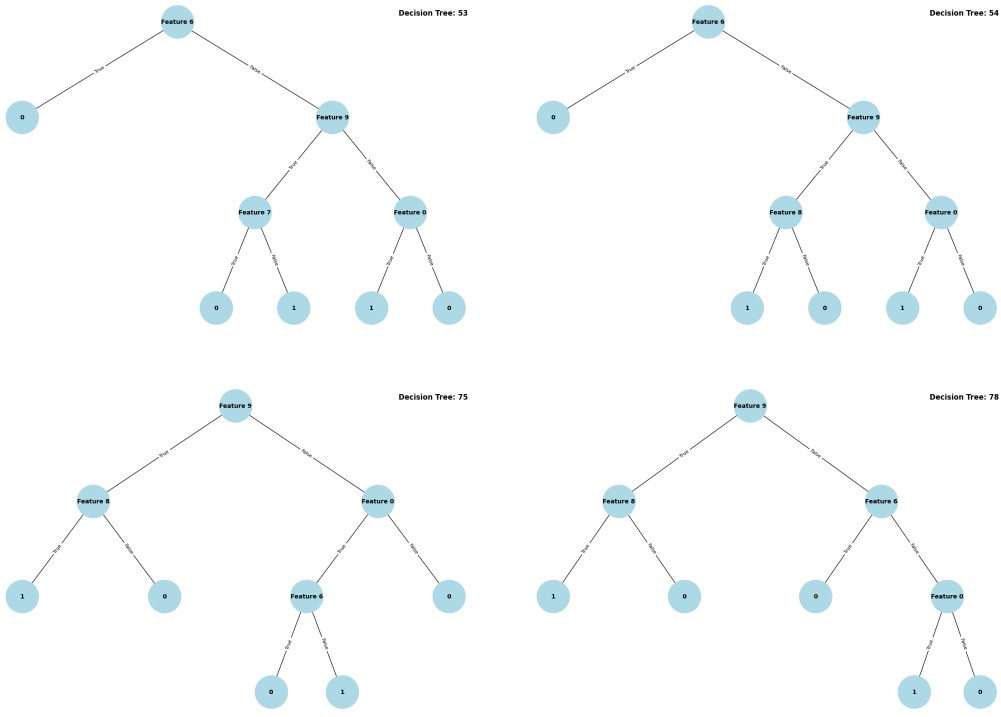

Figure 3: Geometry of the best four decision trees in Group 1 of Figure 1.

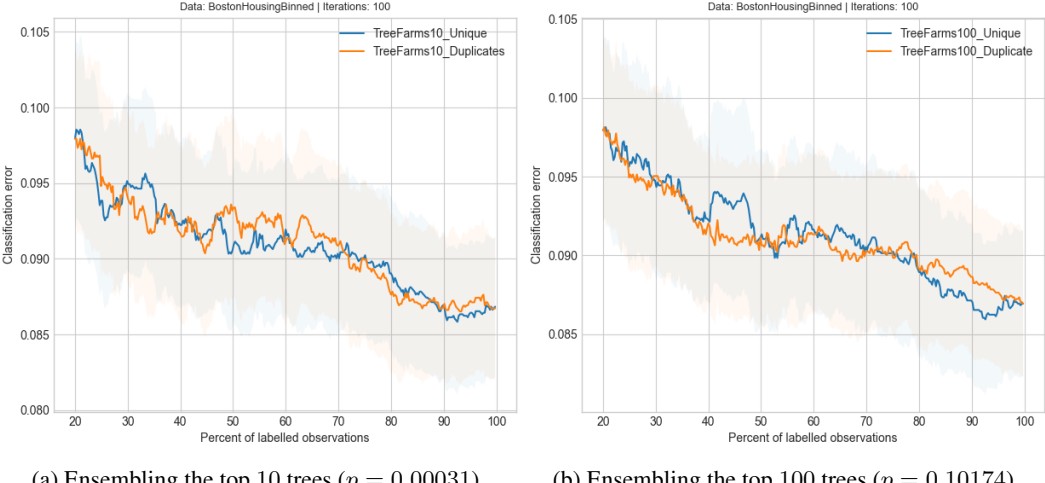

(a) Ensembling the top 10 trees ($p = 0.00031$).

(b) Ensembling the top 100 trees ($p = 0.10174$).

Figure 4

