# OpenReview forum: "Using Rashomon Sets for Robust Active Learning"
_NeurIPS.cc/2024/Workshop/BDU — NeurIPS BDU Workshop 2024 Poster_

### Official Review · Reviewer_vddQ · 2024-09-15
**Interesting idea but lacks compelling experiments.**

**Rating:** 5
**Confidence:** 4

**Review:**

**Summary:** In an active learning setting, instead of considering the prediction from the performing model, the authors propose considering the predictive multiplicity and uncertainty of predictions to choose the observation to label next. They draw an analogy from Economics literature and first develop a Rashomon set of models that include all the models that perform near-optimal. In this set, the authors propose choosing predictions with the highest uncertainty.

---

**Strengths:**
- The idea is interesting.
- The problem statement is clearly motivated.

---

**Weaknesses:**
- Experiments: The experiments reported are basic and could be significantly improved. Although the idea is interesting, the experiments presented do not guarantee it would perform better in real-world settings.
- Writing is basic and could be improved a lot.
- Questions: Some fundamental questions are to be addressed
   - How do you develop a Rashomon set for a real-world ML task? Specifically, how do you develop it for ML tasks, including data corresponding to image/audio/video modalities?
   - As it is not a theoretical contribution, compelling experiments should warrant the efficacy of the proposed method. A naive method may not qualify as a good baseline.

---

### Official Review · Reviewer_hBie · 2024-10-07
**Paper Review: "Using Rashomon Sets for Robust Active Learning"**

**Rating:** 7
**Confidence:** 3

**Review:**

Pros:
Innovative Approach: The paper introduces a novel active learning algorithm, RRAA-AL, which leverages the Rashomon set to account for model ambiguity and predictive multiplicity. This approach is innovative as it goes beyond traditional active learning methods that focus on a single model.
Focus on Model Ambiguity: By focusing on model ambiguity, the paper addresses a significant gap in the active learning literature. It acknowledges that multiple models can explain data well and uses this multiplicity to make more informed decisions about which data points to label.
Robust Methodology: The methodology for selecting the most informative observations by considering the worst-case scenario within a constrained set of plausible models (Rashomon set) is both robust and theoretically sound.
Empirical Validation: The simulation results provide empirical evidence that RRAA-AL can achieve lower training errors compared to traditional active learning approaches, supporting the effectiveness of the proposed method.
Detailed Theoretical Background: The paper provides a thorough overview of Rashomon Theory and its implications for active learning, which helps in understanding the complexity of model ambiguity and its impact on model selection and data labeling.

Cons:
Scalability Concerns: While the paper presents a proof of concept through simulations, the scalability of the approach to real-world scenarios with potentially thousands of models and features remains to be tested.
Dependence on Accurate Model Probability Estimation: The effectiveness of the approach heavily relies on accurately estimating the posterior probabilities of models in the Rashomon set. Any bias or error in these estimations could significantly affect the outcomes.d.

Summary:
The paper "Using Rashomon Sets for Robust Active Learning" presents a compelling new approach to active learning by incorporating model ambiguity and predictive multiplicity through the use of Rashomon sets. This method promises improved robustness and predictive accuracy by considering multiple plausible models. However, challenges related to computational complexity, scalability, and dependency on accurate model probability estimations need further exploration.

---

### Official Review · Reviewer_bXw1 · 2024-10-08
**Generally good**

**Rating:** 7
**Confidence:** 3

**Review:**

The paper tackles an important issue in the domain of Active Learning by addressing model ambiguity using the Rashomon set, which refers to the set of models that are near-optimal but might provide different explanations for the same data. The concept of predictive multiplicity in active learning is novel and crucial, especially when data labeling is costly, and the possibility of several models being equally valid exists.

## Pros
The paper introduces a creative and theoretically sound approach to active learning by focusing on the Rashomon Effect, which hasn't been extensively addressed in prior literature on active learning. It is commendable that the authors move beyond traditional uncertainty-based methods to address the broader problem of model.

The approach of enumerating the Rashomon set of near-optimal models and using posterior model probability to guide the selection of informative observations is innovative. The strategy of focusing on "best worst-case" decisions is a robust and practical contribution to uncertainty modeling.

Active learning is a widely-used method, particularly in fields where labeling data is resource-intensive. Introducing ambiguity-averse decision-making improves the robustness of active learning algorithms, which is a significant advancement for the field. The problem is also well-defined. The authors articulate the issue of predictive multiplicity effectively and convincingly argue for why it poses a critical challenge for existing active learning frameworks.

## Cons
While the theoretical framework is strong, the empirical validation of the proposed approach seems to be somewhat limited. It would benefit from more comprehensive experimentation on diverse datasets to showcase the robustness of the method across different real-world scenarios.

In addition, enumerating the Rashomon set for near-optimal models, while theoretically sound, could be computationally expensive in practice, especially for large datasets or complex models. The paper could have explored the scalability of the approach more thoroughly, including potential optimizations or approximations.

Some parts of the methodology, particularly the algorithmic details, are somewhat dense and could be explained in a more accessible manner for a broader audience. While the target audience is familiar with active learning, more intuitive explanations and examples would enhance the clarity of the approach.

The paper also does not adequately compare its results with traditional active learning methods or other recent advancements that address model uncertainty. A more thorough comparative analysis would make the contribution stand out more convincingly.

---

### Official Review · Reviewer_dxQF · 2024-10-08
**The paper introduces Rashomon sets to address predictive multiplicity and model uncertainty, novel and sound approach but lacks robust empirical validation**

**Rating:** 7
**Confidence:** 4

**Review:**

The paper presents a novel and conceptually sound approach to addressing predictive multiplicity in active learning, which is an important contribution to the field. The work is promising but would benefit from stronger empirical evaluation and a clearer discussion of scalability and practical applicability.

Strengths:
- The paper introduces a novel approach by applying Rashomon sets to active learning, which addresses the issue of predictive multiplicity - where multiple near-optimal models exist that fit the data equally well but offer different predictions. This is an original contribution to the active learning literature.
- The paper opens with a well-defined problem, addressing the issue of predictive multiplicity in active learning. It provides strong motivation for using Rashomon sets to deal with model ambiguity, which is an under-explored area in active learning.
- It effectively tackles the challenge of model ambiguity and uncertainty, which is crucial for ensuring robust decision-making in real-world machine learning applications, especially where model uncertainty can significantly impact outcomes.
- The paper’s use of Rashomon sets is conceptually sound, offering a solution to common issues of model overfitting by considering multiple plausible models instead of relying on a single best model, which is an important step forward in machine learning.

Weaknesses:
Most of the weaknesses listed are around the validation and results
- The simulations are fairly simplistic, demonstrating proof of concept but lacking depth in terms of evaluating the algorithm’s performance on complex, high-dimensional, or noisy data. More rigorous tests are needed to validate the generalizability and robustness of the method.
- Although the paper mentions different metrics Query-By-Committee and uncertainty sampling, it lacks detailed comparative analysis.
- While the paper focuses on near-optimal models within a Rashomon set, it does not explicitly address how the method handles model diversity across different types of models. For instance, how would the method perform if models in the Rashomon set represent very different learning architectures

---

### Decision · Program_Chairs · 2024-10-09

Accept (Poster)